# Solute-Induced Perturbation of the Solvent Microstructure in Aqueous Electrolyte Solutions: Some Uses and Misuses of Structure Making/Breaking Criteria

**Ariel A. Chialvo** [1,*] and **Oscar D. Crisalle** [2]

1   Independent Researcher, Knoxville, TN 37922-3108, USA
2   Department of Chemical Engineering, University of Florida, Gainesville, FL 32611-6005, USA; crisalle@che.ufl.edu
*   Correspondence: ovlaich@gmail.com

**Abstract:** In this article, we raise awareness about the misuses of frequently invoked criteria for structure making/breaking phenomena, resulting from the absence of any explicit cause–effect relationship between the proposed markers and the microstructural perturbation of the solvent induced by the solute. First, we support our assessment with rigorous molecular-based foundations to determine, directly and quantitatively, the solute-induced perturbation of the solvent structure leading to an unambiguous definition of a structure making/breaking event. Then, we highlight and discuss the sources of concealed ambiguities in two of the most frequently invoked structure making/breaking criteria, i.e., Hepler's thermal expansivity-based and Jones–Dole's *B* coefficient-based markers. Finally, we illustrate how the implementation of rigorous molecular-based arguments, in conjunction with the available experimental evidence on a variety of aqueous species at infinite dilution, rule out the validity of these two criteria as structure making/breaking markers and suggest their discontinuation to avoid the perpetuation of myths.

**Keywords:** aqueous electrolyte solutions; solute-induced microstructural perturbation; structure making/breaking events; solution non-idealities; solute–solvent intermolecular asymmetries; Krichevskii parameter; Hepler's criterion; Jones–Dole's *B* coefficient-based criterion

## 1. Introduction

The solvation of a solute in a solvent can be interpreted as the solute-induced (distortion) effect on the original microstructure of the solvent environment leading to the formation of the solvation structure. This distortion can be microstructurally described as a local perturbation of the solvent density originating in the molecular asymmetry between the strength of the original solvent–solvent and the ensuing solute–solvent interactions, thus leading macroscopically to diverse patterns of thermodynamic nonideality [1–3]. Unsurprisingly, researchers have invested significant effort toward the extraction of structural information of electrolyte and nonelectrolyte solutions from scattering experiments, such as X-ray spectroscopy (e.g., EXAFS and XANES) [4–7], neutron scattering with isotope substitution (NDIS) [8–12], hybrid methods involving empirical microstructural refinement of scattering data [13–19], and the interplay between NDIS and molecular simulation [20–26] as means to interpret the underlying links between the intermolecular interaction asymmetries and the resulting thermodynamic behavior.

Alternatively, researchers have used surrogate experimental techniques to study the solvent local environment around a solute in search for connections between the observed macroscopic (thermodynamic) behavior and the microscopic responses detected by different probes leading to the interpretation of the effect of the solute on the microstructure of the solution. However, these experimental tools either provide a rather limited (i.e., short-ranged and/or orientational) view of the solute effects, involve solvent-specific (e.g.,

hydrogen-bonding) approaches, or rely on other types of markers to infer a plausible (yet undefined) microscopic-to-macroscopic link. Two prominent examples of these alternative approaches are the focus of our critical analysis as we introduce them below, i.e., the markers associated with the behavior of the $B$–coefficient in Jones–Dole's equation [27] and the isobaric-thermal expansivity of the solute in solution [28].

On the one hand, soon after Jones and Dole [27] introduced the empirical expression to describe the concentration dependence of the relative viscosity $(\eta/\eta_\alpha^o)$ of dilute solutions in terms of the shear viscosity of the pure $\alpha$–solvent $\eta_\alpha^o$ and the molar concentration $c_\beta$ of the $\beta$–solute in solution, i.e.,

$$(\eta/\eta_\alpha^o) = 1 + Ac_\beta^{0.5} + Bc_\beta \tag{1}$$

followed by the theoretical interpretation of its positive definite $A$–coefficient [29,30], researchers were eager to find an explanation for the behavior of the $B$–coefficient, i.e., the origin of $B \lessgtr 0$ in particular, but not exclusively [31–35], for aqueous ionic solutes. From the early work of Cox and Wolfenden [36], who hypothesized that $B < 0$ could be traced back to "depolymerization" of water as a manifestation of the "rise of structural temperature", according to the theory of Bernal and Fowler available at that time [37], the $B$–coefficient was considered a measure of the solute–solvent interactions. In fact, for the case of aqueous ionic solutes, Wolfenden and collaborators [36,38] advanced the idea of additivity of the anion and cation contributions to the $B$–coefficient of the resulting aqueous salt and, despite the arbitrary nature of the anion–cation division of contributions, the concept was immediately adopted [39–44].

The above effort was likely the first attempt to link the sign of the $B$–coefficient of Jones–Dole's equation to some type of solvent structural motif as a means to explain the experimental observations. In fact, Bernal and Fowler conjectured that the ions affected their solvation water by either loosening or tightening its structure, e.g., a looser structure arising from a weakly solvated ions giving rise to a decrease of the relative viscosity while a tighter structure manifested as a stronger ion solvation translating into an increase of the relative viscosity of the aqueous solution. These ideas were further pursued by Frank and collaborators [45,46], who analyzed the phenomenon according to measures of free volume and "structural entropy" of the solution, and then interpreted through the introduction of the labels "structure-breaking" and "structure-promoting" species.

Furthermore, Tsangaris and Martin [47] introduced an alternative viscosity-based structure making/breaking criterion based on the combinations of signs for the $B$–coefficient and its temperature derivative $(\partial B/\partial T)_P$ by suggesting that "*the sign of $(\partial B/\partial T)_P$ appears to be a more straightforward indicator of structure-breaking or -making ability than sign or size of the B-coefficient*" and that "*a positive $(\partial B/\partial T)_P$ indicates a structure-breaking ion or molecule, and a negative sign, a structure making one*" without any explicit microstructural evidence to support it. Unfortunately, this criterion exacerbates the confusion as we have recently argued [48], given that the resulting four-sign combinations, comprising eight possible structure making/breaking outcomes, are devoid of any cause–effect relationship with the actual microstructural perturbation of the solvent structure around the solute.

On the other hand, early undertakings toward the understanding of the thermodynamic behavior of aqueous solutions were driven by the work of Frank and Evans [46], who proposed the formation of quasi-crystalline hydrogen-bonded structures (icebergs) around highly dilute aqueous nonpolar solutes in order to explain the observed endothermic negative entropy of solution of gases. In other words, this negative entropy of solution was interpreted as an increase of order in the aqueous environment around the solute, or equivalently, a promotor of water structure given that entropy was understood (at that time) as a measure of order. Following these ideas, and supplemented by the experimental evidence of the larger isobaric-thermal expansivity of heavy water over that of light water, Hepler [28] suggested that the thermal expansivity behavior of water was consistent with Frank and Evans' structural model of water, i.e., temperature and pressure increases break down "*structure and cause water to approach "normal" behavior*". Consequently, Hepler promoted

its extension to dilute solutions as a means to interpret the structure-making/breaking behavior of solutes in solution.

In pursuit of this issue, Hepler invoked a Maxwell relation involving the partial molar volumes of the solute at infinite dilution, $\left(\partial \hat{c}^{\infty}_{P_{\beta}}/\partial P\right)_{T} = -T\left(\partial^2 \hat{v}^{\infty}_{\beta}/\partial T^2\right)_{P}$, to keep the analysis at the level of solute–solvent interactions, and assumed that $\left(\partial \hat{c}^{\infty}_{P_{\beta}}/\partial P\right)_{T} > 0$ and, consequently, $\left(\partial^2 \hat{v}^{\infty}_{\beta}/\partial T^2\right)_{P} < 0$, for a structure-breaking solute according to experimental evidence from some hand-picked aqueous systems. As surprising as it might be, and despite the evident lack of explicit microstructural support for the advocated structural marker compounded by the significant ambiguity of the supporting experimental evidence, the thermal expansivity criterion for structure-making/breaking species was adopted and has survived until today [49,50]. Unfortunately, the survival of this criterion is not the result of its accurate predictions, but perhaps for two other factors: its straightforward implementation based on the availability of volumetric data and, more importantly, the absence (until recently) of any rigorous (theoretical) development exposing its invalidity.

Immediately, we can identify a few commonalities between the attempts to link either the solute's *B*–coefficient or the isobaric-thermal expansivity to a structure-making/breaking behavior, including, (i) the chosen systems have been hand-picked to provide a self-consistent outcome by ignoring or overlooking systems not conforming to the preconceived connections, (ii) the structure-making/breaking process, i.e., the microstructural signature of the solvent perturbation around the solute, is vaguely described through a variety of narratives rather than by an explicit microscopic-to-macroscopic unambiguous connection, (iii) the lack of effort to make a direct connection between the actual microstructural behavior of the system and the physical property whose measurement will provide the information to decide the solute's structure-making/breaking ability, and consequently, (iv) the loss of interest in testing the adequacy and/or accuracy of those criteria by confronting their outcomes against precisely-defined model systems for which the answers are precisely known.

In light of this reality, the main goal of this article is to raise awareness of the large community of researchers in physical chemistry and solution chemistry to what can and cannot be construed from two popular and frequently invoked criteria for structure making/breaking phenomena, i.e., Hepler's thermal expansivity formula [28] and the behavior of Jones–Dole's *B*–coefficient [41,47]. More specifically, we would like to emphasize and demonstrate that (i) neither one of the above criteria provides any explicit link between the microstructural perturbation of the solvent caused by the presence of the solute; thus, they suffer from a lack of cause–effect connections between the actual microstructural perturbation and the proposed structure making/breaking marker; (ii) neither criterion can predict the correct solute-induced structural perturbation for the two simplest systems describing either the largest or the smallest solute–solvent intermolecular interaction asymmetry, i.e., those involving the ideal gas *β*–solute in a real *α*–solvent, and the real *β*–solute behaving as an *α*–solvent molecule, systems for which we know precisely the structure making/breaking behavior; and (iii) the macroscopic nature of the above criteria, compounded by the lack of any explicit relationship with the evolution of the solvent microstructure, preclude their reliable use as structure making/breaking markers.

Our presentation is organized as follows: in Section 2, we provide a rigorous molecular-based foundation for the quantitative determination of a solute-induced perturbation of the solvent structure leading to an unambiguous definition of a structure making/breaking event. Then, in Section 3, we highlight the sources of ambiguities of the criteria based on Hepler's isobaric-thermal expansivity and *B*–coefficient markers as descriptors of structure making/breaking events. In Section 4, we illustrate our arguments with experimental evidence for a variety of aqueous electrolyte species at infinite dilution. Finally, we provide some additional thoughts in Section 5.

## 2. Fundamentals Underlying the Description of the Microstructural Perturbation of the Solvent Environment upon Solute Solvation

Our first priority is to provide an unambiguous and precise description of what constitutes a microstructural perturbation of the solvent environment caused by the introduction of a solute species at the prevailing state conditions of the solution, and simultaneously, be able to link rigorously the observed microscopic behavior to the corresponding macroscopic manifestations which will ultimately be the object of the experimental measurements [49]. For that purpose, below, we lay out briefly the main ideas behind the statistical mechanics-based definition of the structure-making/breaking function and discuss the attributes of such a descriptor, including its universality.

### 2.1. What Does Really Mean That a Solute Strengthen/Weaken the Structure of the Solvent?

To address this issue, we invoke the Kirkwood–Buff (KB) fluctuation formalism of mixtures [51] as the statistical mechanical framework able to describe unambiguously the behavior of the system microstructure as volume integrals over pair correlation functions, and connect them to the system's macroscopic chemical and mechanical partial molar properties. In addition to being a rigorous formalism, KB imposes no restrictions to either the nature and type of the intermolecular interaction asymmetry, or the number of components in the system.

The key player here is the KB's total correlation function integral (TCFI), $G_{\beta\alpha}(T, P, x_\beta)$, defined as follows,

$$G_{\beta\alpha}(T, P, x_\beta) \equiv 4\pi \int_0^\infty h_{\beta\alpha}(r) r^2 dr \qquad (2)$$

where $x_\beta$ denotes the mole fraction of the $\beta$–species, while the radial correlation function $h_{\beta\alpha}(r) = g_{\beta\alpha}(r) - 1$ for the $\beta\alpha$–type interactions is defined in terms of the radial distribution functions $g_{\beta\alpha}(r)$ and the ideal gas (IG) uniform distribution $g_{\beta\alpha}^{IG}(r) = 1$ counterpart. Note that the $\alpha$–species density ($\rho_\alpha = \rho x_\alpha$) weighted Equation (2), where $x_\alpha = 1 - x_\beta$, i.e.,

$$\begin{aligned}\mathcal{N}_{\beta\alpha}^{res}(T, P, x_\beta) &= 4\pi \rho x_\alpha \int_0^\infty \left[g_{\beta\alpha}(r) - 1\right] r^2 dr \\ &= \mathcal{N}_{\beta\alpha}(T, P, x_\beta) - \mathcal{N}_\alpha(T, P, x_\beta)\end{aligned} \qquad (3)$$

provides the first rigorous measure of the excess (or its negative counerpart also known as deficit) in the average number of $\alpha$–molecules around any $\beta$–molecule relative to that when the $\alpha$–molecule were uniformly distributed in the system, where $\mathcal{N}_\alpha(T, P, x_\beta) = \rho(1 - x_\beta)V$ and $V$ denotes the volume of the system. Because $\mathcal{N}_{\beta\alpha}(T, P, x_\beta)$ describes the absolute average number of $\alpha$–species around any $\beta$–species at the prevailing $(T, P, x_\beta)$–conditions, then Equation (3) defines the isobaric-isothermal residual $\mathcal{N}_{\beta\alpha}^{res}(T, P, x_\beta)$ counterpart of $\mathcal{N}_{\beta\alpha}(T, P, x_\beta)$, i.e., the effect of the intermolecular interactions on the average number of $\alpha$–molecules around any $\beta$–molecule when the system goes from an ideal gas mixture to the actual mixture of interest.

According to the expressions (2) and (3), we can now focus on the following related quantity,

$$\begin{aligned}\mathcal{N}_{\beta\alpha}^{ex}(T, P, x_\beta) &\equiv \rho x_\alpha \left(G_{\beta\alpha} - G_{\alpha\alpha}\right)_{TPx} \\ &= \left(\mathcal{N}_{\beta\alpha}^{res} - \mathcal{N}_{\alpha\alpha}^{res}\right)_{TPx}\end{aligned} \qquad (4)$$

which, according to the physical meaning of the involved TCFI, $\rho x_\alpha G_{\beta\alpha} = \mathcal{N}_{\beta\alpha}^{res}$, the quantity $\mathcal{N}_{\beta\alpha}^{ex}(T, P, x_\beta)$ represents the average number of $\alpha$–molecules around any $\beta$–solute in excess/deficit to that around any $\alpha$–molecule. Therefore, $\mathcal{N}_{\beta\alpha}^{ex}(T, P, x_\beta)$ becomes a versatile, unambiguous, and direct descriptor of the magnitude of the $\beta$–molecule induced-perturbation of the surrounding $\alpha$–species environment, resulting from the intermolecular asymmetry between the $\alpha\alpha$– and the $\beta\alpha$–intermolecular interactions.

After identifying the $\alpha$–species as the solvent and the $\beta$–species as the solute, immediately, Equation (4) indicates that, as the intermolecular asymmetry between the $\alpha\alpha$– and the

$\beta\alpha$–type interactions vanishes (i.e., the solute behavior becomes identical to that of the solvent), the excess quantity $\mathcal{N}_{\beta\alpha}^{ex}(T, P, x_\beta) \to 0$, as we might expect. However, when the intermolecular asymmetry between the $\alpha\alpha$– and the $\beta\alpha$–type interactions translates into stronger $\beta\alpha$– than $\alpha\alpha$–type interactions, then $G_{\beta\alpha}(T, P, x_\beta) > G_{\alpha\alpha}(T, P, x_\beta)$ and $\mathcal{N}_{\beta\alpha}^{ex}(T, P, x_\beta) > 0$ in Equation (4), which is the manifestation of a strengthening or enhancement of the $\alpha$–solvent environment around the $\beta$–solute. Conversely, when $G_{\beta\alpha}(T, P, x_\beta) < G_{\alpha\alpha}(T, P, x_\beta)$ and $\mathcal{N}_{\beta\alpha}^{ex}(T, P, x_\beta) < 0$ in Equation (4), which is the manifestation of a weakening or depletion of the $\alpha$–solvent environment around the $\beta$–solute. Moreover, we should highlight that an outstanding attribute of Equation (4) is its direct cause–effect connection between the microstructural changes of the mixture and its thermodynamic behavior, a feature that makes it powerful for the linking of the solute–solvent intermolecular interaction asymmetries of the mixture of interest, its microscopic manifestation, and the resulting macroscopic patterns of thermodynamic non-ideality as discussed below.

### 2.2. Need to Provide an Explicit Definition/Criterion for the Structure Making/Breaking Ability of a Solute Species

To decide whether a $\beta$–solute, forming a solution with an $\alpha$–solvent, behaves as a structure making or breaking species, we must first set the molecular-based "meter" criterion that applies equally to any system, regardless of either the type or the magnitude of solute–solvent intermolecular interaction asymmetry [49,52]. In other words, it should apply equally to systems exhibiting either the smallest (i.e., neither making nor breaking such as a solute in a Lewis–Randall ideal solution) [2], the largest (i.e., an ideal gas solute in any real solvent) [3], or any magnitude of solute–solvent molecular interaction asymmetry in between regardless of how these interactions are microscopically described (e.g., hydrogen bonding, electrostatic, multipole, inductive, etc.) [53]. Moreover, the structure making/breaking definition must predict precisely the same answer regardless of the (experimental, theoretical, simulation) probe used in its implementation.

Given that the typical structure making/breaking experimental scenario involves solutes in solutions at infinite dilution, we proceed our analysis of binary systems comprising infinitely dilute solutes at isobaric-isothermal conditions, and define the following structural parameter $\mathcal{S}_{\beta\alpha}^{\infty}(T, P)$ as follows,

$$
\begin{aligned}
\mathcal{S}_{\beta\alpha}^{\infty}(T, P) &\equiv \lim_{x_\beta \to 0} \mathcal{N}_{\beta\alpha}^{ex}(T, P, x_\beta) \\
&= \rho_\alpha^o \left( G_{\beta\alpha}^{\infty} - G_{\alpha\alpha}^{o} \right)_{TP}
\end{aligned}
\tag{5}
$$

so that the magnitude of $\mathcal{S}_{\beta\alpha}^{\infty}(T, P)$ quantifies rigorously and unambiguously how different the microstructure of the $\alpha$–solvent around the $\beta$–solute becomes, relative to that around the $\alpha$–solvent itself, while the sign of $\mathcal{S}_{\beta\alpha}^{\infty}(T, P)$ qualifies the behavior of the $\beta$–solute according to the three possible structural outcomes of the TCFI-difference $\left( G_{\beta\alpha}^{\infty} - G_{\alpha\alpha}^{o} \right)_{TP}$ as follows,

$$
\left( G_{\beta\alpha}^{\infty} - G_{\alpha\alpha}^{o} \right)_{TP} \to
\begin{cases}
G_{\beta\alpha}^{\infty} > G_{\alpha\alpha}^{o} \to \mathcal{S}_{\beta\alpha}^{\infty} > 0 \to structure\text{–}maker \\
G_{\beta\alpha}^{\infty} \cong G_{\alpha\alpha}^{o} \to \mathcal{S}_{\beta\alpha}^{\infty} \cong 0 \to neithermaker\text{–}norbreaker \\
G_{\beta\alpha}^{\infty} < G_{\alpha\alpha}^{o} \to \mathcal{S}_{\beta\alpha}^{\infty} < 0 \to structure\text{–}breaker
\end{cases}
\tag{6}
$$

where the scheme embodied in Equations (5) and (6) provides a one-to-one correspondence between the actual solute-induced perturbation of the solvent microstructure and a meaningful, as well as experimentally accessible, structure-making/breaking function. In fact, by invoking the partial molar volumetric counterpart for the expressions for the TCFI-difference $\left( G_{\beta\alpha}^{\infty} - G_{\alpha\alpha}^{o} \right)_{TP}$, we immediately find the thermodynamic counterpart to Equation (5) that allows a straightforward experimental determination of $\mathcal{S}_{\beta\alpha}^{\infty}(T, P)$, i.e.,

$$
\mathcal{S}_{\beta\alpha}^{\infty}(T, P) = 1 - \left( \hat{v}_\beta^{\infty} / v v_\alpha^{o} \right)_{TP}
\tag{7}
$$

where $\hat{v}_\beta^\infty(T,P)$ and $v_\alpha^o(T,P)$ describe the partial molar volumes of the $\beta$–solute at infinite dilution, $\hat{v}_\beta^\infty(T,P) = \nu\left(v_j^o + G_{jj}^o - G_{ij}^\infty\right)$ with the dissociative stoichiometric coefficient $\nu$, and the corresponding to the pure $\alpha$–solvent at the prevailing state conditions. Note that, because the ion–solvent TCFI's obey the identity $\left(G_{a\alpha}^\infty = G_{c\alpha}^\infty = G_{\beta\alpha}^\infty\right)_{TP}$ [49], the solute induced perturbation of the solvent microstructure is the same for either dissociated species, i.e., $\left(\mathcal{S}_{\beta\alpha}^\infty = \mathcal{S}_{a\alpha}^\infty = \mathcal{S}_{c\alpha}^\infty\right)_{TP}$ where subscripts $a$ and $c$ identify the anion and the cation species, respectively.

At this point, we can shed additional light onto the Kirkwood–Buff integrals involved in the definition of $\mathcal{S}_{\beta\alpha}^\infty(T,P)$, Equation (5), and the resulting microstructural scenarios described by Equation (6). We have already introduced the microscopic (statistical mechanical) definition of the Kirkwood–Buff integral for solvent–solvent interactions, $G_{\alpha\alpha}^o(T,P)$ [51], yet, after we invoke the solvation behavior of an ideal gas $\beta$–solute [54], we can show that $G_{\alpha\alpha}^o(T,P)$ embodies the following distinctive macroscopic meaning,

$$G_{\alpha\alpha}^o = \left(\nu^{-1}\hat{v}_\beta^{\infty,IG\_\beta} - v_\alpha^o\right)_{TP} \tag{8}$$

as the change of partial molar volume of a $\alpha$–solvent molecule in the process of becoming an ideal gas $\beta$–solute at infinite dilution in an environment characterized by the prevailing $(T,P)$ state conditions, where $\hat{v}_\beta^{IG\_\beta}(T,P) = \nu kT\kappa_\alpha^o$ with $\nu = 1$ for non-dissociative solutes [54]. Likewise, we can provide another distinctive macroscopic meaning to the Kirkwood–Buff integral of the solute–solvent interactions, $G_{\beta\alpha}^\infty(T,P)$, i.e.,

$$G_{\beta\alpha}^\infty(T,P) = \nu^{-1}\left(\hat{v}_\beta^{\infty,IG\_\beta} - \hat{v}_\beta^\infty\right)_{TP} \tag{9}$$

as the change of partial molar volume of the real $\beta$–solute at infinite dilution in a real $\alpha$–solvent when the solute–solvent interactions vanish and the species becomes an ideal gas $\beta$–solute.

Alternatively, we could resort to the isothermal-isochoric rate of change of pressure, $\left(\partial P/\partial x_\beta\right)_{T\rho}^\infty$, induced by the $\beta$–solute within the environment of pure $\alpha$–solvent [55], a quantity that plays a crucial role in the understanding of solubility in highly-compressible solvent [2,56] whose finite critical value defines Krichevskii's parameter [57]. Therefore,

$$\mathcal{S}_{\beta\alpha}^\infty(T,P) = -\nu^{-1}\kappa_\alpha^o\left(\partial P/\partial x_\beta\right)_{T\rho}^\infty \tag{10}$$

where $\kappa_\alpha^o$ denotes the isothermal compressibility of the pure $\alpha$–solvent. In fact, the sign of $\left(\partial P/\partial x_\beta\right)_{T\rho}^\infty$ has been pivotal in the characterization of solutes in near-critical solvents, so that according to Equation (10), a $\beta$–solute behaves as a *structure-maker* when $\left(\partial P/\partial x_\beta\right)_{T\rho}^\infty < 0$, and the solute is depicted as non-volatile [58] or attractive [1]. Conversely, a $\beta$–solute behaves as a *structure-breaker* when $\left(\partial P/\partial x_\beta\right)_{T\rho}^\infty > 0$, and the solute is described as volatile [58] or weakly-attractive and repulsive [1] in the jargon of supercritical fluid solutions [2,59].

Incidentally, Equation (10) predicts the divergence of $\mathcal{S}_{\beta\alpha}^\infty(T,P)$ as the state conditions of the pure solvent approach criticality, with the sign of $\left(\partial P/\partial x_\beta\right)_{T\rho}^\infty$, resulting from the divergent behavior of the isothermal compressibility of the solvent $\kappa_\alpha^o(T,P)$. Moreover, while the typical structure making/breaking analysis involves state conditions where $\kappa_\alpha^o(T,P) \lesssim 10^{-5}/P$ (MPa), many novel chemical processes take place in highly compressible media [60–62]. Under these conditions, it becomes advantageous to avoid dealing with divergent quantities, while still capturing the structure making/breaking perturbing effect of the solute. In fact, from the fundamental expression given by Equation (10), we can split $\mathcal{S}_{\beta\alpha}^\infty(T,P)$ into its solvation (i.e., short-range local density perturbation, *SR*) contribution

while isolating its diverging (i.e., long-range or compressibility driven, *LR*) contribution associated with the propagation of the density perturbation as follows [55],

$$\mathcal{S}_{\beta\alpha}^{\infty}(T,P) = \underbrace{-v^{-1}\kappa_{\alpha}^{o,IG}\left(\partial P/\partial x_{\beta}\right)_{T\rho}^{\infty}}_{S_{\beta\alpha}(SR)} \underbrace{-v^{-1}\kappa_{\alpha}^{o,R}\left(\partial P/\partial x_{\beta}\right)_{T\rho}^{\infty}}_{S_{\beta\alpha}(LR)} \tag{11}$$

In Equation (11), we identify $\kappa_{\alpha}^{o,IG} = (\rho_{\alpha}^{o}kT)^{-1}$ as the ideal gas compressibility at the prevailing state conditions, and $\kappa_{\alpha}^{o,R}(T,P) = \kappa_{\alpha}^{o} - \kappa_{\alpha}^{o,IG}$ as the corresponding isobaric-isothermal residual isothermal compressibility. Therefore, from Equations (10) and (11), we immediately find the desired explicit expression for the solvation finite contribution,

$$\mathcal{S}_{\beta\alpha}^{\infty}(SR) = \left(\kappa_{\alpha}^{o,IG}\Big/\kappa_{\alpha}^{o}\right)\mathcal{S}_{\beta\alpha}^{\infty} \tag{12}$$

whose divergent compressibility-driven contribution becomes

$$\mathcal{S}_{\beta\alpha}^{\infty}(LR) = \left(\kappa_{\alpha}^{o,R}\Big/\kappa_{\alpha}^{o}\right)\mathcal{S}_{\beta\alpha}^{\infty} \tag{13}$$

Note also that the isothermal-isochoric rate of change of pressure $\left(\partial P/\partial x_{\beta}\right)_{T\rho}^{\infty}$ can also be written as $\left(\partial P/\partial x_{\beta}\right)_{T\rho}^{\infty} = v\rho_{\alpha}^{o}\left(C_{\alpha\alpha}^{o} - C_{\beta\alpha}^{\infty}\right)\Big/\kappa_{\alpha}^{o,IG}$, where $C_{ij}^{\oplus}(T,P) \equiv 4\pi\int_{0}^{\infty}c_{ij}^{\oplus}(r)r^{2}dr$ defines the direct correlation function integral (DCFI) counterpart of Equation (2) for the *ij*–type of interactions at the prevailing $(T,P)$ state conditions and composition, i.e., $\oplus = o$ for the pure component and $\oplus = \infty$ for the infinite dilution [2]. Thus, after invoking the following macroscopic interpretation for the TCFI [53],

$$C_{\alpha\alpha}^{o}(T,P) = v_{\alpha}^{o}\left[1 - \left(vv_{\alpha}^{o}\Big/\hat{v}_{\beta}^{\infty,IG\_\beta}\right)\right] \tag{14}$$

$$C_{\beta\alpha}^{\infty}(T,P) = v_{\alpha}^{o}\left[1 - \left(\hat{v}_{\beta}^{\infty}\Big/\hat{v}_{\beta}^{\infty,IG\_\beta}\right)\right] \tag{15}$$

which are the counterparts of the TCFI's given by Equations (8) and (9), we obtain

$$\begin{aligned}\left(\partial P/\partial x_{\beta}\right)_{T\rho}^{\infty} &= vkT\rho_{\alpha}^{o}\left(\hat{v}_{\beta}^{\infty} - vv_{\alpha}^{o}\right)\Big/\hat{v}_{\beta}^{\infty,IG\_\beta} \\ &= v\left(\hat{v}_{\beta}^{\infty} - vv_{\alpha}^{o}\right)\Big/\left(\hat{v}_{\beta}^{\infty,IG\_\beta}\kappa_{\alpha}^{o,IG}\right)\end{aligned} \tag{16}$$

$$\begin{aligned}\mathcal{S}_{\beta}^{\infty}(SR) &= -\left(\hat{v}_{\beta}^{\infty} - vv_{\alpha}^{o}\right)\Big/\hat{v}_{\beta}^{\infty,IG\_\beta} \\ &= -\left(\hat{v}_{\beta}^{\infty} - vv_{\alpha}^{o}\right)\Big/\left(vkT\kappa_{\alpha}^{o}\right)\end{aligned} \tag{17}$$

leading straightforwardly to

$$\begin{aligned}\mathcal{S}_{\beta\alpha}^{\infty}(LR) &= \left(\hat{v}_{\beta}^{\infty} - vv_{\alpha}^{o}\right)\Big/\hat{v}_{\beta}^{\infty,IG\_\beta} - \left(\hat{v}_{\beta}^{\infty} - vv_{\alpha}^{o}\right)\Big/vv_{\alpha}^{o} \\ &= \left(\hat{v}_{\beta}^{\infty} - vv_{\alpha}^{o}\right)\left(\left[1\Big/\hat{v}_{\beta}^{\infty,IG\_\beta}\right] - \left[1/vv_{\alpha}^{o}\right]\right)\end{aligned} \tag{18}$$

so that

$$\begin{aligned}\mathcal{S}_{\beta\alpha}^{\infty}(LR) &= (kT\rho_{\alpha}^{o}\kappa_{\alpha}^{o} - 1)\mathcal{S}_{\beta\alpha}^{\infty}(SR) \\ &= -\mathcal{S}_{\beta\alpha}^{\infty,IG\_\beta}\mathcal{S}_{\beta\alpha}^{\infty}(SR)\end{aligned} \tag{19}$$

Equation (19) tells us that the long-range contribution to the structure parameter of any real solute, $\mathcal{S}_{\beta\alpha}^{\infty}(LR)$, becomes proportional to its short-range counterpart $\mathcal{S}_{\beta\alpha}^{\infty}(SR)$ through

the negative value of the structure parameter of the ideal gas $\beta$–solute at the prevailing state conditions, $\mathcal{S}_{\beta\alpha}^{\infty,IG\_\beta}(T,P)$. Consequently,

$$
\begin{aligned}
\mathcal{S}_{\beta\alpha}^{\infty}(T,P) &= \mathcal{S}_{\beta\alpha}^{\infty}(SR) + \mathcal{S}_{\beta\alpha}^{\infty}(LR) \\
&= \left(1 - \mathcal{S}_{\beta\alpha}^{\infty,IG\_\beta}\right)\mathcal{S}_{\beta\alpha}^{\infty}(SR) \\
&= kT\kappa_{\alpha}^{o}\rho_{\alpha}^{o}\mathcal{S}_{\beta\alpha}^{\infty}(SR)
\end{aligned}
\tag{20}
$$

an outcome that confirms the contention that, even for a highly compressible solvent environment, the structure making/breaking behavior of a $\beta$–solute at infinite dilution is still defined by its short range (solvation) contribution. In other words, the isothermal compressibility of the $\alpha$–solvent only magnifies its magnitude by the positive defined factor, $(kT\kappa_{\alpha}^{o}\rho_{\alpha}^{o})$, at the prevailing state conditions.

We can also identify a rigorous connection between $\mathcal{S}_{\beta\alpha}^{\infty}(SR)$ and the corresponding Krichevskii parameter given their common microstructural origin. In fact, from Equation (11) and the definition $\lim\limits_{T,\rho_{\alpha}^{o}\to critical}\left(\partial P/\partial x_{\beta}\right)_{T\rho}^{\infty} \equiv \mathcal{A}_{Kr}$, follows immediately that

$$
\mathcal{A}_{Kr} = -\nu \lim\limits_{T,\rho_{\alpha}^{o}\to critical}\left(\mathcal{S}_{\beta\alpha}^{\infty}\Big/\kappa_{\alpha}^{o}\right)
\tag{21}
$$

and consequently,

$$
\mathcal{A}_{Kr} = -\nu\mathcal{A}_{Kr}^{IG\_\beta}\lim\limits_{T,\rho_{\alpha}^{o}\to critical}\mathcal{S}_{\beta\alpha}^{\infty}(SR)
\tag{22}
$$

In summary, Equations (5)–(9) provide the sought-after rigorous microscopic-to-macroscopic connection that grants an unambiguous description of a $\beta$–solute' propensity to perturb, i.e., to *structure-make/break* the $\alpha$–solvent environment around $\beta$–solute at any state conditions, and leads to the determination of its magnitude based on experimentally-available thermodynamic data, regardless of either the type of solute, solvent, or the nature of the intermolecular interactions. Moreover, it allows for the prediction of the structural response to changes in state conditions and composition of the system, based only on the knowledge of the partial molar volumetric behavior of the species and their $T$– or $P$–derivatives at the original $(T,P,x_{\beta})$–conditions [50].

In fact, we have shown that $\mathcal{S}_{\beta\alpha}^{\infty}(T,P)$ applies equaly to sub-, near-, and super-critical state conditions of the pure $\alpha$–solvent, where the divergent $\mathcal{S}_{\beta\alpha}^{\infty}(T,P)$ becomes unambiguously and rigorous described (Equations (11)–(20)) by its short-ranged and finite (solvation) $\mathcal{S}_{\beta\alpha}^{\infty}(SR)$ counterpart. Most importantly, the $\mathcal{S}_{\beta\alpha}^{\infty}(T,P)$, or $\mathcal{S}_{\beta\alpha}(T,P,x_{\beta})$ for that matter, leads to the rigorous description of the thermodynamic non-ideality behavior of the mixture resulting from the solute–solvent intermolecular asymmetries [3,63]. For instance, we have derived the explicit connections between the structure parameter $\mathcal{S}_{\beta\alpha}^{\infty}(T,P)$ with the solute–solvent intermolecular asymmetry described by $\Delta_{\beta\alpha}^{\infty}(T,P) \equiv G_{\alpha\alpha}^{o} + G_{\beta\beta}^{\infty} - 2G_{\beta\alpha}^{\infty}$ and its associated limiting composition slope of the solute activity coefficient $\left(\partial\ln\gamma_{\beta}^{LR}\Big/\partial x_{\beta}\right)_{TP}^{\infty}$, the Krichevskii parameter $\mathcal{A}_{Kr} = \lim\limits_{T,\rho_{\alpha}^{o}\to crit}\left(-\mathcal{S}_{\beta\alpha}^{\infty}\Big/\kappa_{\alpha}^{o}\right)$, the osmotic second virial coefficients associated with the composition perturbation expansion of the solute chemical potential along four distinctive thermodynamic paths, i.e., $B_{\beta}^{*}(T,\mu_{\alpha})$, $B_{\beta}'(T,P)$, $B_{\beta}''(T,\rho_{\alpha})$, and $B_{\beta}^{\sigma}(T_{\sigma})$, and the resulting patterns of thermodynamic non-ideality behavior [3,49,50,53,64]. Finally, note that for non-dissociative solutes, we simply need to set $\nu = 1$ in the corresponding expressions.

### 3. Critical Analysis of the Ambiguity of Two Widespread Structure Making/ Breaking Markers

The major shortcoming behind the most frequently invoked structure-making/breaking markers is the absolute lack of precision and explicit characterization of what constitutes a structure-making/breaking event, and consequently, how this event might connect to the solute–solvent intermolecular interaction asymmetry of the solution, how it manifests macroscopically in terms of thermodynamic quantities, and how it could be probed/measured experimentally. The solute–solvent interaction asymmetry for a given $\beta$–solute in solution with an $\alpha$–solvent could exhibit a significantly wide span, ranging from that of a non-interacting ideal gas species (i.e., the largest) to that of a species behaving identically to the $\alpha$–solvent (i.e., the absolute smallest) [3], and finally, to either a weakly or a strongly interacting $\beta$–solute, regardless of how we describe the like- and unlike-pair interactions [2].

It becomes immediately obvious that any criterion for the analysis of the *structure making/breaking* process must be able the describe the transition between the structure-making and the structure-breaking perturbations through the crossing of the null-effect boundary, i.e., the condition described by an unperturbed solvent microstructure in the presence of a solute at the prevailing state conditions. To address these issues according to the theoretical developments of Section 2, we discuss below the inadequacy, and consequent failure, of the two most frequently invoked structure making/breaking markers. In particular, we identify and highlight their inability to predict the correct behavior of two precisely-defined model systems, involving the largest and the smallest solute–solvent intermolecular asymmetries, for which we know the exact answer.

#### 3.1. Why Hepler's Isobaric-Thermal Expansivity Criterion Cannot Describe a Structure Making/Breaking Event?

As mentioned in the Introduction, the structure making/breaking ability of an aqueous species has been frequently interpreted in terms of the sign of the isothermal-pressure dependence of its partial molar heat capacity $\left( \partial \hat{c}_{P_\beta}^{\infty} / \partial P \right)_T$, or its more easily accessible Maxwell related expression $-\left( \partial^2 \hat{v}_{\beta}^{\infty} / \partial T^2 \right)_P$, as suggested by Hepler [28], who proposed the criterion based on the conjecture that "*since increasing pressure would also break up the bulky aggregates, the same reasoning suggests that the heat capacity of pure water should decrease with increasing pressure*" this behavior could be extended to species in solution at infinite dilution. The fact that there is no cause–effect relation between either $\left( \partial \hat{c}_{P_\beta}^{\infty} / \partial P \right)_T$ or $-T\left( \partial^2 \hat{v}_{\beta}^{\infty} / \partial T^2 \right)_P$ and the magnitude (and/or sign) of the solute-induced perturbation of the solvent microstructure [49] makes futile any attempt to interpret the structure making/breaking events from either volumetry or pressure perturbation calorimetry (PPC) [65].

As we have discussed previously [49,50], Hepler's criterion fails two fundamental requirements stemming from its inability to describe the structure making/breaking behavior of systems comprising either: (a) a $\beta$–solute species behaving identically to an $\alpha$–solvent species resulting in an unperturbed solvent microstructure, or (b) the solvation of a non-interacting ideal gas $\beta$–solute in a real $\alpha$–solvent, which represents the largest possible perturbation of the solvent microstructure. Consequently, Hepler's criterion cannot account for the structure making-to-structure breaking transition with changes in the state conditions of the system.

Beyond the questionable assumptions underlying this criterion [66], we can immediately: (i) demonstrate the failure of this thermal expansivity-based marker to predict the correct structure making/breaking behavior for the ideal gas $\beta$–solute, *IG_$\beta$*, in a real $\alpha$–solvent, and that for the $\beta$–solute when behaving identically as the real $\alpha$–solvent, as well as (ii) identify the isobaric-volumetric behavior of the pure $\alpha$–solvent that would be required by the expansivity-based criterion to be obeyed.

On the one hand, for the largest solute–solvent intermolecular asymmetry, we have the $IG\_\beta$ solute, which according to Equation (7) and the identity $\hat{v}_\beta^{\infty,IG\_\beta}(T,P) = vkT\kappa_\alpha^o$ [54,67], leads to the exact form for its structure parameter [49]

$$\mathcal{S}_{\beta\alpha}^{\infty,IG\_\beta}(T,P) = (1 - kT\rho_\alpha^o \kappa_\alpha^o) \tag{23}$$

whose behavior leads to the following scheme,

$$\mathcal{S}_{\beta\alpha}^{\infty,IG\_\beta}(T,P) \begin{cases} > 0 \to \textit{structure maker} \to \kappa_\beta^{IG\_\beta} > \kappa_\alpha^o \\ = 0 \to \textit{unperturbed structure} \to \kappa_\beta^{IG\_\beta} = \kappa_\alpha^o \\ < 0 \to \textit{structure breaker} \to \kappa_\beta^{IG\_\beta} < \kappa_\alpha^o \end{cases} \tag{24}$$

The essential points from Equation (24) are three-fold: (i) the $IG\_\beta$ in a real $\alpha$–solvent at ambient conditions will exhibit a structure-making behavior, (ii) the solvent microstructure stays unperturbed along the states where the solvent behaves as an ideal gas fluid, satisfying the $\kappa_\alpha^o = [kT\rho_\alpha^o]^{-1}$ condition, and (iii) after crossing this boundary, the ideal gas solute becomes a structure-breaker. Note also that, according to Equation (10), and the fact that $\lim_{P,T\to critical}(\partial P/\partial x_\beta)_{T\rho}^{\infty,IG\_\beta} = \rho_{\alpha,c}^o kT_c$ [56], $\mathcal{S}_{\beta\alpha}^{\infty,IG\_\beta}(T_c,P_c) \to -\infty$, indicating that the critical conditions of the $\alpha$–solvent is located on the left side of the curve representing $\mathcal{S}_{\beta\alpha}^{\infty,IG\_\beta}(T,P) = 0$ or its equivalent $\left(\kappa_\alpha^o = \kappa_\beta^{IG\_\beta}\right)_{T\rho_\alpha^o}$, as illustrated in Figures 8 and 9 in ref. [3] for the solvent water and the Lennard–Jones fluid, respectively.

The issue of interest here concerns the inability of the isobaric-thermal expansivity-based criterion to predict the transition across the curve $\mathcal{S}_{\beta\alpha}^{\infty,IG\_\beta}(T,P) = 0$, as described by $-T\left(\partial^2\hat{v}_\beta^{\infty,IG\_\beta}/\partial T^2\right)_P = 0$ in Hepler's terms. In fact, we find that $\left(\partial^2\hat{v}_\beta^{\infty,IG\_\beta}/\partial T^2\right)_P^{\kappa_\alpha^o = \kappa_\beta^{IG\_\beta}} \neq 0$, in other words, Hepler's criterion cannot describe the structure making-to-breaking transition for the ideal gas solute $IG\_\beta$ in a real $\alpha$–solvent along the $\left(\kappa_\alpha^o = \kappa_\beta^{IG\_\beta}\right)$–line given that

$$\left(\partial^2\hat{v}_\beta^{\infty,IG\_\beta}/\partial T^2\right)_P = k\left[2\left(\partial\kappa_\alpha^o/\partial T\right)_P + T\left(\partial^2\kappa_\alpha^o/\partial T^2\right)\right]^{\kappa_\alpha^o = \kappa_\beta^{IG\_\beta}} \neq 0 \tag{25}$$

after considering that $\left(\partial\kappa_\alpha^{o,IG}/\partial T\right)_P^{\mathcal{S}_{\alpha\beta}^{\infty,IG\_\beta}=0} = \left(\partial^2\kappa_\alpha^{o,IG}/\partial T^2\right)_P^{\mathcal{S}_{\alpha\beta}^{\infty,IG\_\beta}=0} = 0$ with $\left(\partial\kappa_\alpha^o/\partial T\right)_P^{\mathcal{S}_{\alpha\beta}^{\infty,IG\_\beta}=0} \neq 0$ and $\left(\partial^2\kappa_\alpha^o/\partial T^2\right)_P^{\mathcal{S}_{\alpha\beta}^{\infty,IG\_\beta}=0} \neq 0$.

On the other hand, for the smallest solute–solvent intermolecular asymmetry as characterized by the special case of a Lewis–Randall ideal solution, $LR - IS$, and regardless of the definition of structure making/breaking we choose, the outcome must be a null microstructural perturbation depicted here by $\mathcal{S}_{\beta\alpha}^{\infty,LR-IS}(T,P) = 0$ in the schematic Equation (6). While the physical representation for the null solute–solvent intermolecular asymmetry is characterized by $\left(G_{\alpha\alpha}^o + G_{\beta\beta}^\infty - 2G_{\beta\alpha}^\infty\right)_{TP} = 0$ [68,69], its simplest and most advantageous case occurs when the Kirkwood–Buff integrals obey $\left(G_{\alpha\alpha}^o = G_{\beta\beta}^\infty = G_{\beta\alpha}^\infty \neq 0\right)_{TP}$ condition, not only resulting in the microstructural signature of a pure $\alpha$–solvent [67], but also leading to the prototypical fingerprint for the microstructural transition between structure-making and structure-breaking processes driven by the solute–solvent intermolecular interaction asymmetry. This general and rigorous condition for the existence of a boundary between structure-making and structure breaking scenarios affords another opportunity to test the validity of the thermal expansivity-based criterion to predict this transition, which in Hepler's terms becomes described by $\left(\partial^2\hat{v}_\beta^\infty/\partial T^2\right)_P = \left(\partial^2 v_\alpha^o/\partial T^2\right)_P$.

Typically, pure fluids at normal (around ambient) conditions exhibit $T\left(\partial^2 v_\alpha^o/\partial T^2\right)_P > 0$, suggesting that for these fluids to describe the boundary $\left(\partial^2 v_\alpha^o/\partial T^2\right)_P = 0$ line, they would require meeting at least two rather unusual properties: an isobaric thermal expansivity $\beta_{T\alpha}^o$ independent of the state conditions, i.e., $\left(\partial \ln v_\alpha^o/\partial T\right)_P \neq \mathcal{F}(T, P)$, and a linear temperature-dependent isothermal compressibility, i.e., $\kappa_\alpha^o(T, P) \equiv \mathcal{G}(P) + \mathcal{K}(P)T$. We are not aware of any pure solvent following this highly unlikely type of behavior; therefore, we must conclude that Hepler's criterion cannot possibly describe (let alone predict) the structure-making to structure-breaking transition events involving common solvents (especially water). In other words, the conjectured Hepler's criterion cannot account for the solute-induced perturbation of the solvent microstructure of infinitely dilute solution comprising extreme solute–solvent intermolecular asymmetries, i.e., either the smallest (zero, for the $LR - IS$) or the largest (for the $IG\_\beta$ solute).

### 3.2. Why the Behavior of the Jones–Dole's B-Coefficient Cannot Be Taken as a Structure Making/Breaking Marker?

The Jones–Dole's equation for the relative (to pure solvent) viscosity of the dilute solution is represented by Equation (1); for the current analysis, it becomes advantageous to recast it as follows,

$$
\begin{aligned}
\eta_r\left(T, P, c_\beta\right) - 1 \;&= A c_\beta^{0.5} + B c_\beta \text{ as } c_\beta \to 0 \\
&\cong \ln \eta_r\left(T, P, c_\beta\right)
\end{aligned}
\tag{26}
$$

where $\eta_r = \eta/\eta_j^o$ with $\eta_\alpha^o \equiv \eta\left(c_\beta = 0\right)$, $c_\beta$ denotes the molar concentration of the $\beta$–solute, while the coefficients $A$ and $B$ account for the direct ion–ion and the (ion) solute–solvent interactions, respectively [27]. The Jones–Dole's equation has been empirically built, with an intentionally introduced [70] $c_\beta^{0.5}$–composition dependence to obey the Debye–Hückel limiting behavior when dealing with electrolyte solutes, provides a theoretical interpretation for the $A$–coefficient [30,71], and predicts $A > 0$ for all electrolytes [43,72], while $A = 0$ for non-electrolyte solutes.

If we accepted the Jones–Dole's equation to be an accurate representation of the shear viscosity of a dilute solution, and because the $A$–coefficient has been derived around the Debye–Hückel limiting behavior, the $B$–coefficient should also follow from the corresponding isothermal-isobaric composition limiting slope, i.e.,

$$
\begin{aligned}
B \;&= \lim_{c_\beta \to 0}\left[\left(\eta_r - 1 - A c_\beta^{0.5}\right)\middle/ c_\beta\right] \\
&\cong \lim_{c_\beta \to 0}\left[\left(\ln \eta_r - A c_\beta^{0.5}\right)\middle/ c_\beta\right]
\end{aligned}
\tag{27}
$$

where the experimental evidence indicates that $B \gtrless 0$. Moreover, because the $B$–coefficient becomes the pre-factor behind the linear concentration dependence in the Jones–Dole's equation, this coefficient has been associated with the solvent-mediated solute–solute interactions, and consequently, with a conjectured structure marker, namely: $B > 0$ for a structure-maker species and $B < 0$ for a structure-breaker species [40–42,46], later supplemented with the alternative isobaric-temperature derivative $\left(\partial B/\partial T\right)_P$ [47].

The common feature between the two viscosity-based structure making/breaking markers mentioned above is the absolute lack of any explicit cause–effect link between either the $B$–coefficient, or its temperature derivative $\left(\partial B/\partial T\right)_P$, and the actual solute-induced perturbation of the solvent microstructure [48]. For that reason, we would like to provide a few observations and draw plausible connections between the experimental evidence on the behavior of the Jones–Dole's $B$–coefficient and the precisely-defined structure making/breaking parameter $\mathcal{S}_{\beta\alpha}^\infty(T, P)$. Given the empirical nature of Jones–Dole's equation, we need to find whether or not the $B$–coefficient contains any embedded microstructural information, and also to connect this microstructural information to the alleged struc-

ture making/breaking makers based on the signs of both $B$–coefficient [40,42,46,73–82] and its isobaric-temperature derivative $(\partial B/\partial T)_P$ [47,83–100].

One plausible connection between the experimental evidence of the Jones–Dole's $B$–coefficient and a precisely-defined microstructural behavior of the $\beta$–solute at infinite dilution can be drawn by following Feakins et al.'s [101] transition state (TS)-based interpretation of the $B$–coefficient. In fact, Feakins et al. invoked the transition state theory [102] to derive an expression for the $B$–coefficient of a $\beta$–solute at infinite dilution in an $\alpha$–solvent as follows (highlights of its derivation are given in Appendix A below),

$$B = \beta v_\alpha^o \left(\Delta \mu_\beta^{\natural,\infty} - \nu \Delta \mu_\alpha^{\natural,o}\right) - \left(\hat{v}_\beta^\infty - \nu v_\alpha^o\right) \tag{28}$$

with $\nu = \nu^+ + \nu^-$ for an electrolyte, i.e., $\nu = 1$ for non-dissociative solutes, where $\Delta \mu_\beta^{\natural,\oplus}$ and $\hat{v}_\beta^\oplus$ denote the molar Gibbs free energy of activation of viscous flow and the corresponding partial molar volume for the $\beta$–species, respectively, at the $\oplus$–composition condition, i.e., either an infinite dilution or a pure component.

Immediately, we find that the transition-state interpretation of the $B$–coefficient, as described by Equation (28), comprises two contributions, i.e., the $\left(\hat{v}_\beta^\infty - \nu v_\alpha^o\right)$ term as the solute-induced volumetric effect on the solvent structure, and its activation Gibbs free energy counterpart for the viscous flow, $\left(\Delta \mu_\beta^{\natural,\infty} - \nu \Delta \mu_\alpha^{\natural,o}\right)$. Obviously, the volumetric $\left(\hat{v}_\beta^\infty - \nu v_\alpha^o\right)$ term provides a direct link to the structure making/breaking parameter $\mathcal{S}_{\beta\alpha}^\infty(T,P)$ as discussed in Section 2.2, i.e.,

$$B = v_\alpha^o \left[\nu \mathcal{S}_{\beta\alpha}^\infty + \beta\left(\Delta \mu_\beta^{\natural,\infty} - \nu \Delta \mu_\alpha^{\natural,o}\right)\right] \tag{29}$$

which suggests that, under Feakins et al.'s framework, the $B$–coefficient comprises some information about the solute-induced effect on the solvent microstructure. However, even if we assumed the reliability of the Jones–Dole's representation for the composition dependent relative viscosity of the solution, we cannot in principle take the sign of either the $B$–coefficient or its temperature-derivative $(\partial B/\partial T)_P$ counterpart as a marker of the structure making/breaking nature of the dilute $\beta$–solute because the $B$–coefficient, and consequently $(\partial B/\partial T)_P$, involve the $\left(\Delta \mu_\beta^{\natural,\infty} - \nu \Delta \mu_\alpha^{\natural,o}\right)$ term that also contributes to the sought sign.

To support and illustrate this contention, we resort again to the system involving an ideal gas $\beta$–solute ($IG\_\beta$) in a real $\alpha$–solvent, $\Delta \mu_\beta^{\natural,\infty,IG\_\beta} = 0$ [102] and $\mathcal{S}_{\beta\alpha}^{\infty,IG\_\beta} = (1 - kT\rho_\alpha^o \kappa_\alpha^o)$ from Equation (23), so that Equation (29) becomes

$$\begin{aligned} B^{IG\_\beta} &= v_\alpha^o \left(\mathcal{S}_{\beta\alpha}^{\infty,IG\_\beta} - \beta \Delta \mu_\alpha^{\natural,o}\right) \\ &= v_\alpha^o \left[1 - \beta \Delta \mu_\alpha^{\natural,o} - \left(\kappa_\alpha^o/\kappa_\alpha^{IG}\right)\right] \end{aligned} \tag{30}$$

In this equation, we have $\beta \Delta \mu_\alpha^{\natural,o} = \ln(\eta_\alpha^o v_\alpha^o/h\mathcal{N})$ [102], where $\eta_\alpha^o$ and $v_\alpha^o$ denote the shear viscosity and molar volume of the pure $\alpha$–solvent, while $h$ and $\mathcal{N}$ identify Planck and Avogadro constants, respectively. According to Equation (28), the $B^{IG\_\beta}$–coefficient depends exclusively on the thermodynamic properties of the pure $\alpha$–solvent at the prevailing state conditions. Therefore, we can explore the phase diagram of the $\alpha$–solvent to find the $\rho_\alpha^o - T$ conditions where $\mathcal{S}_{\beta\alpha}^{\infty,IG\_\beta}(T,\rho_\alpha^o)$, and simultaneously $B^{IG\_\beta}(T,\rho_\alpha^o)$, to predict a structure-making or breaking behavior. For this task, we already know the location of the line $\mathcal{S}_{\beta\alpha}^{\infty,IG\_\beta}(T,\rho_\alpha^o) = 0$, where the isothermal compressibility of the solvent becomes equal to that of an ideal gas at the same state conditions, i.e., $\kappa_\alpha^o(T,\rho_\alpha^o) = (kT\rho_\alpha^o)^{-1}$ [3,50]. In

what follows, we illustrate the lack of a one-to-one correspondence between the sign of $\mathcal{S}_{\beta\alpha}^{\infty,IG\_\beta}(T,\rho_\alpha^o)$ and $B^{IG\_\beta}(T,\rho_\alpha^o)$.

For this purpose, let us start by selecting the structure-making behavior, $\mathcal{S}_{\beta\alpha}^{\infty,IG\_\beta} = 1 - kT\rho_\alpha^o\kappa_\alpha^o > 0$ [49], that leads to either $\eta_\alpha^o v_\alpha^o < h\mathcal{N}$ (i.e., $\beta\Delta\mu_\alpha^{\natural,o} < 0$) or $\eta_\alpha^o v_\alpha^o > h\mathcal{N}$ (i.e., $\beta\Delta\mu_\alpha^{\natural,o} > 0$ with $0 < \beta\Delta\mu_\alpha^{\natural,o} < \mathcal{S}_{\beta\alpha}^{\infty,IG\_\beta}$). From a thermodynamic viewpoint, this condition means also that $\beta\Delta\mu_\alpha^{\natural,o} + kT\rho_\alpha^o\kappa_\alpha^o < 1$ and will result in a $B^{IG\_\beta}(T,\rho_\alpha^o) > 0$ in (sign) agreement with the starting condition $\mathcal{S}_{\beta\alpha}^{\infty,IG\_\beta}(T,\rho_\alpha^o) > 0$. However, when $\eta_\alpha^o v_\alpha^o > h\mathcal{N}$ but $\beta\Delta\mu_\alpha^{\natural,o} > \mathcal{S}_{\beta\alpha}^{\infty,IG\_\beta}$ then, $\beta\Delta\mu_\alpha^{\natural,o} + kT\rho_\alpha^o\kappa_\alpha^o > 1$ and $B^{IG\_\beta}(T,\rho_\alpha^o) < 0$, in contrast (opposite sign) to the starting condition $\mathcal{S}_{\beta\alpha}^{\infty,IG\_\beta}(T,\rho_\alpha^o) > 0$.

Likewise, if we choose the structure-breaking behavior, $\mathcal{S}_{\beta\alpha}^{\infty,IG\_\beta} = 1 - kT\rho_\alpha^o\kappa_\alpha^o < 0$ [49], then, when $\eta_\alpha^o v_\alpha^o < h\mathcal{N}$ (i.e., $\beta\Delta\mu_\alpha^{\natural,o} < 0$ and therefore, $\beta\left|\Delta\mu_\alpha^{\natural,o}\right| > \left|\mathcal{S}_{\beta\alpha}^{\infty,IG\_\beta}\right|$), we obtain $B^{IG\_\beta}(T,\rho_\alpha^o) > 0$ whose thermodynamic meaning, $1 - kT\rho_\alpha^o\kappa_\alpha^o > \beta\Delta\mu_\alpha^{\natural,o} < 0$ with $kT\rho_\alpha^o\kappa_\alpha^o > 1$, leads to a $B^{IG\_\beta}(T,\rho_\alpha^o) > 0$ in contrast (opposite sign) to the starting condition $\mathcal{S}_{\beta\alpha}^{\infty,IG\_\beta}(T,\rho_\alpha^o) < 0$. However, when either $\eta_\alpha^o v_\alpha^o > h\mathcal{N}$ or $\eta_\alpha^o v_\alpha^o < h\mathcal{N}$ (i.e., $\beta\Delta\mu_\alpha^{\natural,o} < 0$ or $\beta\left|\Delta\mu_\alpha^{\natural,o}\right| < \left|\mathcal{S}_{\beta\alpha}^{\infty,IG\_\beta}\right|$), then we find that $\beta\Delta\mu_\alpha^{\natural,o} + kT\rho_\alpha^o\kappa_\alpha^o > 1$ and $B^{IG\_\beta}(T,\rho_\alpha^o) < 0$, in (sign) agreement with the starting $\mathcal{S}_{\beta\alpha}^{\infty,IG\_\beta}(T,\rho_\alpha^o) < 0$.

The preceding rigorous microscopic-to-macroscopic analysis associated with the behavior of the ideal gas $\beta$–solute in a real $\alpha$–solvent illustrates the lack of uniqueness of the structure making/breaking marker based on the sign of the $B$–coefficient. In fact, this criterion would not be able to distinguish a structure-making, $\mathcal{S}_{\beta\alpha}^{\infty,IG\_\beta} > 0$, from a structure-breaking, $\mathcal{S}_{\beta\alpha}^{\infty,IG\_\beta} < 0$, behavior since either $B^{IG\_\beta} < 0$ or $B^{IG\_\beta} > 0$ could simultaneously describe a $\mathcal{S}_{\beta\alpha}^{\infty,IG\_\beta} > 0$ and a $\mathcal{S}_{\beta\alpha}^{\infty,IG\_\beta} < 0$ scenario depending on the state conditions of the pure $\alpha$–solvent.

Obviously, we can follow the same argument to analyze the microstructural behavior of a real binary mixture whose $B$–coefficient is described by Equation (28). Indeed, if the system exhibited $\mathcal{S}_{\beta\alpha}^{\infty} > 0$, then $B > 0$ would represent a structure-making behavior whenever $\mathcal{S}_{\beta\alpha}^{\infty} > \left(v\Delta\mu_\alpha^{\natural,o} - \Delta\mu_\beta^{\natural,\infty}\right)\big/vkT$, otherwise, the structure-making condition $\mathcal{S}_{\beta\alpha}^{\infty} > 0$ would be described by $B < 0$ as long as $0 < \mathcal{S}_{\beta\alpha}^{\infty} < \left(v\Delta\mu_\alpha^{\natural,o} - \Delta\mu_\beta^{\natural,\infty}\right)\big/vkT$. However, if the system showed $\mathcal{S}_{\beta\alpha}^{\infty} < 0$, then $B < 0$ would describe a structure-breaking behavior whenever $\mathcal{S}_{\beta\alpha}^{\infty} < \left(v\Delta\mu_\alpha^{\natural,o} - \Delta\mu_\beta^{\natural,\infty}\right)\big/vkT$, otherwise, the structure breaking condition $\mathcal{S}_{\beta\alpha}^{\infty} < 0$ would require a $B > 0$. As before, due to the lack of a one-to-one correspondence between the sign of the structure making/breaking parameter $\mathcal{S}_{\beta\alpha}^{\infty}$ and that of the $B$–coefficient, the ad hoc assumption about the sign of the Jones–Dole's $B$–coefficient cannot be taken as a marker of structure making/breaking trends as has been traditionally done in the literature [40,42,46,73–82].

Finally, we provide a brief comment on the ability of the $\left(\partial B/\partial T\right)_P$ counterpart to discriminate between structure-making and structure-breaking solutes. We should note that, regardless of the theoretical description for the behavior of the $B$–coefficient, there are in principle four possible combinations of pair conditions $B \gtrless 0$ and $\left(\partial B/\partial T\right)_P \gtrless 0$ for a binary mixture, resulting in eight structure making/breaking scenarios whose thorough analysis has been presented elsewhere [48]. According to the arguments above, and those in SI-C of the Supporting Information in [48], we determine that neither $\left(\partial B/\partial T\right)_P$ alone nor the $\left[B, \left(\partial B/\partial T\right)_P\right]$ pair-combination can offer an unambiguous correspondence between their signs and the structure making/breaking nature of the $\beta$–solute as described here by the molecular-based parameter $\mathcal{S}_{\beta\alpha}^{\infty}$.

## 4. Experimental Evidence of Structure Making/Breaking Behavior in Aqueous Electrolytes and Comparison against Predictions from the Conjectured Markers

In what follows, we illustrate the theoretical developments of Section 2 and findings of Section 3 based on irrefutable experimental evidence from a variety of aqueous solutes at infinite dilution for which we have available volumetric and rheologic experimental data.

### 4.1. Illustration of the Behavior of the Short- and Long-Range Contributions to the Structure Parameter along the Liquid Branch of the Coexistence Phase Envelope of Water

For that purpose, we selected a few simple aqueous electrolyte $\beta$–solutes and invoked the accurate SOCW thermodynamic model [103,104] to describe the behavior of their partial molar volume at infinite dilution $\hat{v}_\beta^\infty \left( \rho_{H_2O}^o \right)_\sigma$, as described by Equation(A11) of Appendix B, using the parameterization from Table 4 of [103], along the liquid branch of the water phase envelope. These infinite dilute aqueous systems include the following $\beta$–solutes: $NaCl$, $NaBr$, $NaI$, $NaCH_3CO_2$ ($NaAc$), $LiCl$, $KCl$, $CsCl$, $NH_4Cl$, and $NaOH$.

In Figures 1 and 2, we plot the orthobaric ($\sigma$) density dependence of the short-range contribution to the structure making/breaking parameter, $\mathcal{S}_{\beta H_2O}^\infty(SR)$ (described by Equation (A13) in Appendix B), in comparison with its full value counterpart, $\mathcal{S}_{\beta H_2O}^\infty \left( \rho_{H_2O}^o \right)_\sigma$ (described by Equation (A12) in Appendix B), and identify the resulting finite $\mathcal{S}_{\beta H_2O}^{\infty,critical}(SR)$ quantity associated to the corresponding Krichevskii parameter via Equation (22). On the one hand, Figures 1 and 2 illustrate the divergent nature of $\mathcal{S}_{\beta H_2O}^\infty \left( \rho_{H_2O}^o \right)_\sigma$ as the orthobaric solvent density approaches its critical point given that $\hat{v}_\beta^\infty \left( \rho_{H_2O}^o \to \rho_{H_2O,crit}^o \right)_\sigma \sim \kappa_{H_2O}^o$, and consequently, $\mathcal{S}_{\beta H_2O}^\infty \left( \rho_{H_2O}^o \to \rho_{H_2O,crit}^o \right)_\sigma \to +\infty$ (right ordinate). On the other hand, the corresponding $\mathcal{S}_{\beta H_2O}^\infty(SR)$ exibits a finite critical limit (left ordinate), which defines the Krichevskii parameter $\mathcal{A}_{Kr}$ of the $\beta$–solute in water according to Equation (22). Note that, because $\mathcal{A}_{Kr}^{IG-\beta} = k \left( T\rho_{H_2O}^o \right)_{crit}$, these figures also indicate that all these electrolyte solutes describe $\mathcal{A}_{Kr} < 0$, i.e., these electrolyte solutes behave as attractive [1] or non-volatile species [58].

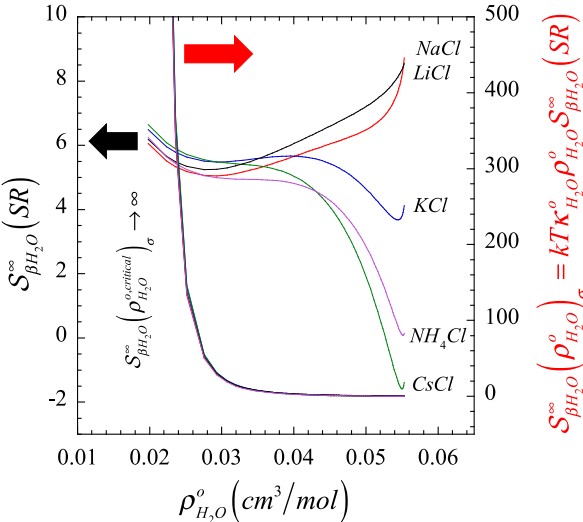

**Figure 1.** Color-coordinated comparison of the behavior of the liquid-phase orthobaric density dependence for the structure making/breaking parameter $\mathcal{S}_{\beta H_2O}^\infty \left( \rho_{H_2O}^o \right)_\sigma$, and its short-ranged counterpart $\mathcal{S}_{\beta H_2O}^\infty(SR)$, of a few aqueous univalent cation chlorides where we highlight (right) the divergent nature of the parameter at the critical point of the solvent, and (left) the corresponding finiteness of their short-range counterpart associated with the Krichevskii parameter.

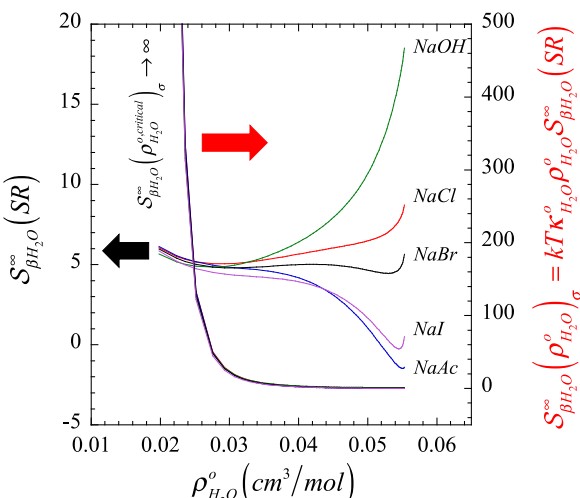

**Figure 2.** Color-coordinated comparison of the behavior of the liquid-phase orthobaric density dependence for the structure making/breaking parameter $\mathcal{S}_{\beta H_2O}^{\infty}\left(\rho_{H_2O}^{o}\right)_{\sigma}$, and its short-ranged counterpart $\mathcal{S}_{\beta H_2O}^{\infty}(SR)$, of a few aqueous univalent sodium salts where we highlight (right) the divergent nature of the parameter at the critical point of the solvent, and (left) the corresponding finiteness of their short-range counterpart associated with the Krichevskii parameter.

### 4.2. Comparison between the Predictions of the Two Common Structure Making/Breaking Markers and the Actual Microstructural Behavior

Here, we illustrate the incompatibility between the microstructural behavior and the volumetric, as well as rheological, data of a variety of dilute aqueous solutions and the predictions from the structure making/breaking discussed in Sections 2 and 3. These systems cover a wide range of solute–solvent intermolecular interaction asymmetries, and comprise organic as well as inorganic electrolyte solutes almost exclusively at ambient conditions. In Table 1, we display the reported volumetric experimental data in the form of the temperature-factored isobaric-temperature derivative $-T\left(\partial^2 \hat{v}_{\beta}^{\infty} \big/ \partial T^2\right)_P$ of the infinitely dilute solute in water as Hepler's structure making/breaking marker, and the structural parameter $\mathcal{S}_{\beta H_2O}^{\infty}(T, P)$ according to Equation (7) as the descriptor of the actual measure of the solute-induced microstructural perturbation. The rheologic data involve the *B*–coefficient of the Jones–Dole's equation and its isobaric temperature derivative $\left(\partial B / \partial T\right)_P$ as reported from the composition dependence of the shear viscosity of the solutions, via regression of the Jones–Dole's equation. For comparison purposes, we also included the predicted behavior from the systems involving the ideal gas solute, *IG_β*, and the *β*–solute behaving like a $H_2O$–molecule (*LR − IS*) [3].

Table 1 illustrates the disparity of the results from the ad hoc criteria based on either the behavior of the isobaric-thermal expansivity or the *B*–coefficient and its isobaric-temperature derivative $\left(\partial B / \partial T\right)_P$, as compared against the actual microstructural responses of the systems accounted by $\mathcal{S}_{\beta H_2O}^{\infty}(T, P)$. In fact, the comparison between column 2 and columns 3–5 indicates that $-T\left(\partial^2 \hat{v}_{\beta}^{\infty} \big/ \partial T^2\right)_P$ as well as the *B*–coefficient and either its temperature derivative or their combination not only fail to predict unambiguously the structure making/breaking nature of the solutes, but also reveal, unsurprisingly, a broad inconsistency between them. This lack of consistency between the thermal expansivity- and the viscosity-based markers highlights the lack of one-to-one (i.e., uniqueness of the) connection between the solute-induced perturbation of the solvent microstructure and the markers.

**Table 1.** Experimental structure making/breaking parameter $\mathcal{S}^{\infty}_{\beta H_2O}(T, P)$ for aqueous infinitely dilute solutes at ambient conditions in comparison with predictions based on Hepler's isobaric-thermal expansivity marker as well as Jones–Dole's $B-$ coefficient and $(\partial B/\partial T)_P$ derivative criteria.

| $\beta$–Solute | $\mathcal{S}^{\infty}_{\beta H_2O}(T,P)$ [a] | $(\partial B/\partial T)_P$ [b] | $B$ [b] | $-T(\partial^2 \hat{v}^{\infty}_{\beta}/\partial T^2)_P$ | Ref. |
|---|---|---|---|---|---|
| Water (LR-IS) | 0 | 0 | 0 | maker | This work |
| Ideal gas | maker | <0 | <0 | breaker | This work |
| creatine | breaker | <0 | >0 | maker | [105] |
| creatinine | breaker | <0 | >0 | maker | [105] |
| nicotinic acid | breaker | <0 | >0 | maker | [106] |
| l-ascorbic acid | breaker | >0 | >0 | breaker | [50,94,106] |
| glycine | breaker | >0 | >0 | breaker | [107] |
| alanine | breaker | >0 | >0 | breaker | [107] |
| DTAB [c] | breaker | >0 | <0 | breaker | [107] |
| l-serine | breaker | <0 | >0 | maker | [96] |
| l-arginine | breaker | <0 | >0 | maker | [96] |
| choline-biotinate | breaker | <0 | >0 | maker | [108] |
| choline-nicotinate | breaker | <0 | >0 | maker | [108] |
| choline-ascorbate | breaker | >0 | >0 | maker | [108] |
| LiCy [d] | breaker | <0 | >0 | breaker | [109,110] |
| NaCy [d] | breaker | ~0 | >0 | maker | [109,110] |
| KCy [d] | breaker | >0 | >0 | maker | [109,110] |
| CaCl$_2$ | maker | >0 | >0 | breaker | [111,112] |
| CdCl$_2$ | maker | <0 | >0 | breaker | [113,114] |
| NiCl$_2$ | maker | >0 | <0 | breaker [e] | [114,115] |
| NH$_4$NO$_3$ | breaker | >0 | <0 | breaker | [116,117] |
| MgCl$_2$ | maker | <0 | >0 | breaker | [114,118] |

[a] Defined to Equation (7); [b] See Equation (1); [c] Dodecyl-trimethyl-ammonium bromide; [d] Alkaline metal cyclohexyl sulfamate; [e] According to our 3rd-order polynomial regression of the $\hat{v}^{\infty}_{\beta}(T)$ data from Herrington et al. [113].

Indeed, the lack of uniqueness or complete ambiguity becomes clearly exposed as follows: either (a) from Table 1, we could choose four aqueous systems comprising structure-making solutes, $\mathcal{S}^{\infty}_{\beta H_2O}(T, P) > 0$, such as $\beta = (CaCl_2, NiCl_2, CdCl_2, IG\_\beta)$ and observe that their structure making/breaking ability are described by four different $[B, (\partial B/\partial T)_P]$–sign combinations, while Hepler's criterion describes all four solutes as structure-breakers; or (b) from Tables 1 and 2, we could choose four aqueous systems involving structure-breaking solutes, $\mathcal{S}^{\infty}_{\beta H_2O}(T, P) < 0$, such as $\beta = (glycine, DBTA, LiCy, IG\_\beta)$ and find that their structure making/breaking ability are described again by four different $[B, (\partial B/\partial T)_P]$–sign combinations, while Hepler's criterion describes the first three solutes as structure-breakers and the supercritical $IG\_\beta$ solute as a structure-maker.

**Table 2.** Representative systems illustrating the eight $[B, (\partial B/\partial T)_P]$ pair combinations and their resulting structure making/breaking parameter $\mathcal{S}^{\infty}_{\beta H_2O}(T, P)$ for infinitely dilute aqueous solutes at ambient conditions.

| Aqueous Solute | $(\partial B/\partial T)_P$ | $B$ | $\mathcal{S}^{\infty}_{\beta H_2O}(T,P)$ |
|:---:|:---:|:---:|:---:|
| $CaCl_2$ | >0 | >0 | >0 (maker) |
| choline-ascorbate | >0 | >0 | <0 (breaker) |
| $CdCl_2$ | >0 | <0 | >0 (maker) |
| $NH_4NO_3$ | >0 | <0 | <0 (breaker) |
| $MgCl_2$ | <0 | >0 | >0 (maker) |
| LiCy [a] | <0 | >0 | <0 (breaker) |
| IG_β | <0 | <0 | >0 (maker) |
| IG_β [b] | <0 | <0 | <0 (breaker) |

[a] Alkaline metal cyclohexyl sulfamate; [b] Supercritical conditions where $\kappa^o_{H_2O} > \kappa^{o,IG}_{H_2O}$.

The lack of uniqueness in the structure making/breaking markers defined around the behavior of the *B*–coefficient becomes even more obvious in Table 2, where we identify four representative pairs of infinitely dilute *β*–solutes in which each pair displays precisely the same particular behavior for the *B*–coefficient, while the individual members of the pair exhibit opposite structure making/breaking parameter $\mathcal{S}^{\infty}_{\beta\alpha}(T, P)$. For example, $CaCl_2$ and choline-ascorbate share the same $\left[B, \left(\partial B/\partial T\right)_P\right] > 0$ behavior, although $\mathcal{S}^{\infty}_{CaCl_2\ H_2O} > 0$ while $\mathcal{S}^{\infty}_{Choline\ Ascorbate\ H_2O} < 0$.

The preceding assessment of the experimental evidence in conjunction with the rigorous definition of $\mathcal{S}^{\infty}_{\beta\alpha}(T, P)$, or $\mathcal{S}_{\beta\alpha}\left(T, P, x_\beta\right)$ for that matter, highlights the unreliability of the predictions from the two widespread structure making/breaking markers resulting from the lack of explicit microstructure-to-macroscopic relations in their definitions to confer an unambiguous description of the propensity of a *β*–solute to distort the microstructure of the *α*–solvent. It also emphasizes the fact that the sought unambiguous description according to Equation (7) only requires two pieces of volumetric information, namely the partial molar volumes of the pure *α*–solvent and of the *β*–solute at infinite dilution. For those readers eager to jump directly to the application of the parameter $\mathcal{S}^{\infty}_{\beta\alpha}(T, P)$ as a fundamentally-based tool for the description and measurement of the magnitude of a solute's ability to perturb the solvent structure, we provide in Appendix C the step-by-step procedure towards its straightforward calculation.

## 5. Final Remarks and Outlook

We must emphasize that our analysis does not judge the accuracy or the usefulness of the composition and temperature dependencies of either the viscosity coefficients or the volumetric behavior of dilute solutions, but rather assesses the validity of their microstructural interpretation based on either the resulting viscosity *B*–coefficient, and corresponding temperature derivative, or the isobaric thermal expansivity markers. Researchers have often been susceptible to adding a name to a phenomenon under investigation as if by inserting a label the phenomenon becomes intuitively understood. This practice has frequently led to misunderstanding and confusion, as we have repeatedly witnessed [2,119] during the early attempts to gain understanding of the solubility enhancement of sparingly soluble solutes in highly compressible solvents, a subject that bears striking similarities with the arguments of the present work. Indeed, the evolving, vague narrative intended to aid the microstructural interpretation of the alluded solvation phenomenon took a variety of names, including solvent clustering [120], densification/cavitation [121], density augmentation/depletion [122], and molecular charisma [123], which led to controversies resulting from the lack of precision in the meaning of "local or short-ranged effect, drastic or significant microstructural changes" [124,125]. The past events suggest that we should refrain from inserting a struc-

ture maker/breaker label (also known as kosmotropic/chaotropic) [52] to a solute species until we fully understand what that phenomenon means, by focusing on addressing the real issues, e.g., what we expect to learn about the thermodynamic behavior of the system by analyzing its microstructural behavior. In the words of Richard Feynman, [126] we must recognize " ... *the difference between knowing the name of something and knowing something*".

In this work, we raised awareness of what can and cannot be inferred from Hepler's thermal expansivity and Jones–Dole's *B*–coefficient criteria as structure making/breaking markers, including: (i) neither one provides cause–effect connections between the actual microstructural perturbation and the proposed markers; (ii) neither criterion can predict the correct structure making/breaking answer for the two simplest systems describing either the largest or the smallest solute–solvent intermolecular interaction asymmetry, systems for which we know precisely the structure making/breaking behavior; and (iii) the macroscopic nature of the above criteria, compounded by the lack of any explicit link to the evolution of the solvent microstructure, preclude their reliable use as structure making/breaking markers and, as such, their use should be discontinued to avoid the perpetuation of confusion.

**Author Contributions:** Conceptualization, A.A.C.; methodology, A.A.C.; formal analysis, A.A.C.; resources, O.D.C.; writing—original draft preparation, A.A.C.; writing—review and editing, O.D.C. All authors have read and agreed to the published version of the manuscript.

**Funding:** This research received no external funding.

**Institutional Review Board Statement:** Not applicable.

**Informed Consent Statement:** Not applicable.

**Data Availability Statement:** Not applicable.

**Acknowledgments:** The authors express their gratitude to Josef Sedlbauer, Technical University Liberec (Czechia), for kindly commenting on the predictive ability of the SOCW equation used in this work to describe the orthobaric behavior of the partial molar volumes aqueous alkali halides at infinite dilution.

**Conflicts of Interest:** The authors declare no conflict of interest.

## Appendix A. Transition State Interpretation of Jones–Dole's B-Coefficient

We must first highlight the approximations supporting Feakins et al. [101] transition-state interpretation of the *B*–coefficient. For that purpose, we first recast Jones–Dole's equation [27] as follows,

$$\ln(\eta/\eta_\alpha^o) \cong A c_\beta^{0.5} + B c_\beta \qquad (A1)$$

where $\eta/\eta_\alpha^o - 1 \cong \ln(\eta/\eta_\alpha^o)$ and then invoke Glasstone–Laidler-Eyring's [102] transition state expressions for the viscosity of a pure $\alpha$–solvent, $\eta_\alpha^o$,

$$\eta_\alpha^o = (h\mathcal{N}/v_\alpha^o) \exp\left(\beta \Delta G_\alpha^{\natural,o}\right) \qquad (A2)$$

and that of the dilute solution comprising a $\beta$–solute, $\eta$,

$$\eta = (h\mathcal{N}/v) \exp\left(\beta \Delta G_{\beta,\alpha}^{\natural}\right) \qquad (A3)$$

where $h$ and $\mathcal{N}$ denote Planck and Avogadro constants, while $v_\alpha^o$ and $\Delta G_\alpha^{\natural,o}$ describe the pure solvent molar volume and the corresponding Gibbs free energy of activation for the viscous flow process, respectively, while $v$ and $\Delta G_{\beta,\alpha}^{\natural}$ denote the corresponding molar volume of the resulting solution and the average Gibbs free energy of activation for the viscous flow of the components in solution.

Equations (A1)–(A3) allow connecting the two Jones–Dole coefficients with the change of Gibbs free energy of activation for the viscous flow process and the relative (to the pure solvent) molar volume of the dilute solution, i.e.,

$$Ac_\beta^{0.5} + Bc_\beta \cong \ln(v_\alpha^o/v) + \beta\left(\Delta G_{\alpha,\beta}^\natural - \Delta G_\alpha^{\natural,o}\right) \tag{A4}$$

and

$$Ac_\beta^{0.5} + Bc_\beta \cong \ln(v_\alpha^o/v)_{c_\beta\to 0} + \beta\left(\Delta G_{\beta,\alpha}^\natural - \Delta G_\alpha^{\natural,o}\right)_{c_\beta\to 0} + \mathcal{F}\left(c_\beta^{0.5}\right) \tag{A5}$$

where $\mathcal{F}\left(c_\beta^{0.5}\right)$ will be identified below. Moreover, after recalling that $v = v^+ + v^-$ for a general electrolyte solution, and $v = 1$ for a non-electrolyte solution, we have that

$$\begin{aligned}\Delta G_{\beta,\alpha}^\natural &= x_\beta\Delta\mu_\beta^\natural + x_\alpha\Delta\mu_\alpha^\natural \\ &= x_\beta\left(\Delta\mu_\beta^\natural - v\Delta\mu_\alpha^\natural\right) + \Delta\mu_\alpha^\natural\end{aligned} \tag{A6}$$

$$\begin{aligned}v &= x_\beta\hat{v}_\beta + x_\alpha\hat{v}_\alpha \\ &= x_\beta\left(\hat{v}_\beta - v\hat{v}_\alpha\right) + \hat{v}_\alpha\end{aligned} \tag{A7}$$

which lead to the following approximation of the relative molar volume,

$$\begin{aligned}\ln(v/v_\alpha^o)_{x_\beta\to 0} &\cong -\ln\left[1 - x_\beta\left(v\hat{v}_\alpha - \hat{v}_\beta\right)/v_\alpha^o\right]_{x_\beta\to 0} \\ &\cong c_\beta\left(vv_\alpha^o - \hat{v}_\beta^\infty\right)\end{aligned} \tag{A8}$$

where $c_\beta = x_\beta/v$ defines the molar concentration of the $\beta$–solute. Consequently, from Equations (A5)–(A8), we obtain,

$$\left(\Delta G_{\beta,\alpha}^\natural - \Delta G_\alpha^{\natural,o}\right)_{c_\beta\to 0} = c_\beta v_\alpha^o\left(\Delta\mu_\beta^{\natural,\infty} - v\Delta\mu_\alpha^{\natural,o}\right) \tag{A9}$$

which provides the identification of $\mathcal{F}\left(c_\beta^{0.5}\right) = Ac_\beta^{0.5}$ in A5 and the TS-interpretation of the $B$–coefficient in the Jones–Dole equation as follows,

$$B = \left(vv_\alpha^o - \hat{v}_\beta^\infty\right) + \beta v_\alpha^o\left(\Delta\mu_\beta^{\natural,\infty} - v\Delta\mu_\alpha^{\natural,o}\right) \tag{A10}$$

A direct comparison between A10 and Equation (18) in the original derivation of [101] highlights that these authors assumed $v = 1$ even when analyzing electrolyte solutions, a feature that has evaded the attention of many authors and might have contributed to errors in the calculation of transition state viscosity-related quantities and their interpretation in the literature. While the error introduced in the case of a dissociative solute comprising $v = 2$ is about 5–6% [96,109,127–129], depending on the relative ratio $\left(v_\alpha^o/\hat{v}_\beta^\infty\right)$, it becomes significantly larger, i.e., about 20–25% for dissociative solutes comprising $v = 4$.

### Appendix B. Structure Making/Breaking Parameter from the SOCW Representation of the Partial Molar Volumes of Simple Electrolyte Solutes

We invoke the Sedlbauer–O'Connell–Wood (SOCW) [103,130] expression for the partial molar volume of the $(v^+:v^-)$ ions conforming a $\beta$–electrolyte solute at infinite dilution in the $\alpha$–solvent, i.e.,

$$\begin{aligned}\hat{v}_i^\infty(T,P) = &\{(1 - z_i) + a_i\rho_\alpha^o - d_i + b_i\rho_\alpha^o[\exp(\vartheta\rho_\alpha^o) - 1] + \\ &\delta\rho_\alpha^o[\exp(\lambda\rho_\alpha^o) - 1] + c_i\rho_\alpha^o\exp(\theta/T)\}kT\kappa_\alpha^o + d_i/\rho_\alpha^o\end{aligned} \tag{A11}$$

where the $i$–subindex identifies the individual ion bearing the electrostatic charge $z_i$, with the regressed parameters $[a_i, b_i, c_i, d_i]$ given in Table 4 in [103], the universal constants

$\theta = 1500\text{K}$, $\vartheta = 0.005$ m$^3$/kg, and $\lambda = -0.01$ m$^3$/kg, while $\delta = 0$ for cations and $\delta = -0.645$ m$^3$/kg for anions. While these parameters have been regressed from high-pressure liquid phases, they describe accurately the orthobaric behavior of electrolytes at infinite dilution [131]. Then, considering that the partial molar volume of the $\beta$–solute at infinite dilution equals to $\hat{v}_\beta^\infty(T,P) = \nu^+ \hat{v}_+^\infty + \nu^- \hat{v}_-^\infty$, the structure making/breaking parameter from Equation (7) with $\nu^+ = \nu^- = 1$ and $\nu = 2$ can be described by the following expression,

$$\mathcal{S}_{\beta\alpha}^\infty(T,P) = 1 - 0.5d_\pm - 0.5\{ \quad 2 - d_\pm + \rho_\alpha^o a_\pm + \rho_\alpha^o b_\pm [\exp(\vartheta\rho_\alpha^o) - 1] + \\ \delta\rho_\alpha^o[\exp(\lambda\rho_\alpha^o) - 1] + \rho_\alpha^o c_\pm \exp(\theta/T)\}kT\kappa_\alpha^o\rho_\alpha^o \tag{A12}$$

where $a_\pm = a_+ + a_-$, $b_\pm = b_+ + b_-$, $c_\pm = c_+ + c_-$ and $d_\pm = d_+ + d_-$. Consequently, from Equation (20) we find that

$$\mathcal{S}_{\beta\alpha}^\infty(SR) = \kappa_\alpha^{o,IG}(1 - 0.5d_\pm)/\kappa_\alpha^o - 0.5\{ \quad 2 - d_\pm + \rho_\alpha^o a_\pm + \rho_\alpha^o b_\pm[\exp(\vartheta\rho_\alpha^o) - 1] + \\ \delta\rho_\alpha^o[\exp(\lambda\rho_\alpha^o) - 1] + \rho_\alpha^o c_\pm \exp(\theta/T)\} \tag{A13}$$

## Appendix C. Practical Guide to the Calculation of the Fundamentally-Based Structure Making/Breaking Marker $\mathcal{S}_{\beta\alpha}^\infty(T,P)$

For that purpose, we assume that we have available the isobaric-isothermal composition (molar concentration $c_\beta$ or any alternative) dependence of either the molar volume of the dilute solution ($v(c_\beta)$) or the apparent molar volume of the dilute $\beta$–solute in an $\alpha$–solvent ($v_\beta^\varphi(c_\beta)$) and proceed as follows:

1.  Calculate the partial molar volume of the $\beta$–solute at infinite dilution $\hat{v}_\beta^\infty$ as the composition limiting behavior $\hat{v}_\beta^\infty = \lim_{c_\beta \to 0} v_\beta^\varphi(c_\beta)$;

2.  Calculate the partial molar volume of the pure $\alpha$–solvent, i.e., $v_\alpha^o = \lim_{c_\beta \to 0} v(c_\beta)$;

3.  Calculate the structure making/breaking parameter $\mathcal{S}_{\beta\alpha}^\infty(T,P) = 1 - \left(\hat{v}_\beta^\infty/\nu v_\alpha^o\right)_{TP}$, Equation (7), after considering the stoichiometric $\nu$–parameter of the $\beta$–solute, either $\nu = \nu^+ + \nu^-$ for an ionic solute or $\nu = 1$ for a non-dissociative solute;

4.  Compare the ratio $\left(\hat{v}_\beta^\infty/v_\alpha^o\right)_{TP}$ with the stoichiometric $\nu$–parameter:

    a.  If $\left(\hat{v}_\beta^\infty/v_\alpha^o\right)_{TP} < \nu$, then $\mathcal{S}_{\beta\alpha}^\infty(T,P) > 0$, i.e., the $\beta$–solute behaves as a structure making at the prevailing state conditions;

    b.  If $\left(\hat{v}_\beta^\infty/v_\alpha^o\right)_{TP} > \nu$, then $\mathcal{S}_{\beta\alpha}^\infty(T,P) < 0$, i.e., the $\beta$–solute behaves as a structure breaking at the prevailing state conditions;

    c.  If $\left(\hat{v}_\beta^\infty/v_\alpha^o\right)_{TP} \cong \nu$, then $\mathcal{S}_{\beta\alpha}^\infty(T,P) \cong 0$, i.e., the $\beta$–solute induces a negligible structure perturbation at the prevailing state conditions.

Note that these outcomes are completely independent on the nature of the solvent and the type of the solute–solvent intermolecular interactions, i.e., the $\mathcal{S}_{\beta\alpha}^\infty(T,P)$ applies equally to aqueous and non-aqueous solvents, electrolyte and non-electrolyte solutions, and require no information whatsoever about any (solvent-specific or otherwise) interaction mechanism such as hydrogen bonding.

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
