# Peer review of "Solute-Induced Perturbation of the Solvent Microstructure in Aqueous Electrolyte Solutions: Some Uses and Misuses of Structure Making/Breaking Criteria"

_liquids, doi:10.3390/liquids2030008_

Round 1

Reviewer 1 Report

Solute-induced Perturbation of the Solvent Microstructure in Aqueous Electrolyte Solutions

by

Ariel A. Chialvo and Oscar D. Crisalle

This work is an excellent review of definition and (mis)uses of a structure making/breaking criteria. First, a rigorous molecular-based foundation for the quantitative determination of a solute-induced perturbation of the solvent structure leading to an explicit definition/criterion for the structure making/breaking ability of a solute species. Next, the criteria based on Hepler’s isobaric-thermal expansivity and Jones-Dole’s B coefficient are critically discussed. It is strongly advised not to use them as structure making/breaking markers.

It is strongly stressed, that the paper is not a critic to the experimental data (without them, this paper would not be possible), but clearly shows their wrong interpretation.

The application of the parameter Sba¥ (T,P).as a fundamentally-based tool for the description and measurement  of the magnitude of a solute’s ability to perturb the solvent structure is advised. In appendix C the step-by-step procedure towards straightforward calculation of this parameter is given. Thus, this paper is of big importance and has to be published.

There is only one concern: the quality of figures must be proved-some used fonts (axis titles)  seem to be to big and not in accordance with others (in legends, numbers,…).

Author Response

We have resized the figures and revised the notation in the legends to be consistent with that in the figure axes

Reviewer 2 Report

attached
